# Association between demographic, clinical characteristics and severe complications by SARS-CoV-2 infection in a community-based healthcare network in Chile

Javiera Leniz[1], Sam Hernández-Jaña[2], Mauricio Soto[3], Eduardo Arenas[4], Paula Margozzini[1,5], Francisco Suarez[6], Daniel Capurro[7], María Paulina Rojas[3], Claudia Bambs[1,5,8]*

1 Escuela de Salud Pública, Facultad de Medicina, Pontificia Universidad Católica de Chile, Santiago, Chile, 2 IRyS Group, Physical Education School, Pontificia Universidad Católica de Valparaíso, Valparaíso, Chile, 3 Departamento de Medicina Familiar, Escuela de Medicina, Pontificia Universidad Católica de Chile, Santiago, Chile, 4 Unidad de Gestión de Informática, Ancora UC, Santiago, Chile, 5 Center for Cancer Prevention and Control, Fondap 152220002, Faculty of Medicine, Pontificia Universidad Católica de Chile, Santiago, Chile, 6 Departamento de Estadística, Análisis y Gestión de la Información en Salud, Servicio de Salud Metropolitano Sur-Oriente, Santiago, Chile, 7 School of Computing and Information Systems, University of Melbourne, Melbourne, Australia, 8 Advanced Center for Chronic Diseases, Fondap 151300, Faculty of Medicine, Pontificia Universidad Católica de Chile, Santiago, Chile

* cbambs@uc.cl

## Abstract

### Background

Most of the evidence on risk factors for COVID-19 complications comes from North America or Europe with very little research from Latin-America. We aimed to evaluate the association between sociodemographic, clinical factors and the risk of COVID-19 complications among adults in Chile, the fifth Latin-American country with more COVID-19 reported cases since de beginning of the Pandemic.

### Methods

A retrospective population-based cohort study using data from electronic health records from a large Primary Care Network, linked to national hospital, immunization, Covid-19 PCR surveillance, mortality and birth records. We included people 18+ years old enrolled in the Primary Care Network between 1st January 2020 and 31st December 2021. Using Multivariate Cox proportional hazard models, we evaluate the association between sociodemographic, clinical characteristics with three COVID-19 complications: (1) a hospital admission, (2) an ICU admission, and (3) death due to a COVID-19 infection that occurred between the 1st January 2020 and the 31st December 2021.

### Results

44,674 people were included. The mean age was 44.30 (sd 17.31), 55.6% were female, 15.9% had a type of healthcare insurance for people from the lowest category of income, 11.6% and 9.4% had a record of hypertension or diabetes mellitus diagnosis. Among the

**Data Availability Statement:** Data may be obtained from a third party and are not publicly available. The data that support the findings of this study are available from the Servicio de Salud Metropolitano Sur-Oriente and the Ancora Network but restrictions apply to the availability of these data, which were used under license for the current study, and so are not publicly available. The Department of Statistics at the SSMSO holds all the information regarding patients that are enrolled in primary healthcare centres in their area, including administrative and clinical records from primary care and hospital care. The SSMSO act as data controller of the dataset. A formal application process and a data sharing agreement with a Chilean institution is needed in order to access the data. More information about the SSMSO and the Department of Statistics can be found here https://redsalud.ssmso.cl/. Information for data access can be requested at deagis@ssmso.cl. The Unidad de Informatica (UGI) at the Ancora Network manages all clinical and administrative records of the three primary healthcare centres related to the Network. The UGI act as data controller of the dataset. A formal application process and a data sharing agreement with a Chilean institution is needed in order to access the data. More information about the Ancora Network and the UGI can be found here https://www.ancorauc.cl/. Information for data access can be requested at cisauc@ucchristus.cl.

**Funding:** The author(s) received no specific funding for this work.

**Competing interests:** The authors have declared that no competing interests exist.

44,674 people, 455 (1.02%) had a hospital admission due to a COVID-19 infection and 216 (0.48%) of them also had an ICU admission. Among the 44,674 people,148(0.33%) died due to COVID-19 infection. Older age and male sex were consistently associated with a higher risk of the three COVID-19 complications. Hypertension and diabetes were associated with a higher risk of a hospital admission and death, but not with an ICU admissions due to COVID-19 infection. Having two or more COVID-19 vaccine doses compared with no doses was associated with a lower risk of any hospital admission (HR 0.81; 95% CI 0.77–0.84), an ICU admission (HR 0.60; 95% CI 0.57–0.63) and death (HR 0.50; 95% CI 0.46–0.54). Pregnant or puerperal women were more likely to be admitted to hospital (HR 2.89; 95% CI 1.41–5.89) or ICU (HR 3.04; 95% CI 1.01–9.14).

## Conclusions

Sociodemographic and clinical factors associated with COVID-19 complications such as age, sex and pre-existing conditions were comparable to those reported in similar studies from higher-income countries, and can be used to predict severity in COVID-19 patients.

## Introduction

The COVID-19 pandemic had an enormous impact in terms of morbidity and mortality globally [1]. It has been estimated that by the end of 2021, 3.39 billion people had been infected, which represents the 43.9% of the global population [2]. By December 2021 a total of 5.94 million COVID-19 deaths had been reported worldwide, but more than 18 million deaths due to COVID were estimated based on excess mortality measures [3]. While Latin-American and Caribbean was among the regions with a lower rate of COVID-19 infections, it was among those with the highest cumulative rate of COVID-19 deaths [2], likely to be explained by less prepared healthcare systems and social security policies [4]. The severity of the disease caused by the COVID-19 virus could range from mild symptoms to an acute respiratory syndrome leading to hospital admissions, ICU admissions and death [5–7].

Several recent systematic reviews have summarised the literature regarding risk factors for COVID-19 complications [8–12]. Results from these systematic reviews show that factors consistently associated with a higher risk of COVID-19 complications are pre-existing comorbidities, such as hypertension, heart failure, diabetes mellitus, coronary heart disease, cancer, chronic obstructive pulmonary disease, smoking, age and male gender [8, 9, 12]. Obesity, psychiatric conditions and pregnancy have also been associated to COVID-19 negative outcomes and severe symptoms [5, 13–18]. Nevertheless, most of the evidence in these systematic reviews comes from Europe, USA and Asia, with very little research on Latin-American populations [8, 9, 12]. More recently, a large observational multinational study explored risk factors for COVID-19 complications among hospitalized patients in Latin America [19]. Risk factors independently associated with progression to ICU admission were age, shortness of breath, and obesity [19]. Another large population-based study among confirmed-COVID-19 cases in Mexico, age was the most predictive factor for mortality. Renal disease, hypertension, diabetes and obesity were as well associated with a higher risk of dead due to a COVID-19 infection [20]. Nevertheless, studies exploring risk factors for COVID-19 complications at population level are scarce.

Chile has a well-established social security system, with 98% of the population having health insurance, 77% of them affiliated to a public insurance system (FONASA) [21]. Despite being

ranked as the 28th country with the highest Global Health Security (GHS) Index [22], and developed an integrated reporting system for COVID-19 surveillance [23], Chile was the fifth Latin-American country with more COVID-19 reported cases since the beginning of the Pandemic, and a cumulative number of 5.3 million COVID-19 infections and 61 thousand deaths by November 2023 [24]. Understanding to what extent factors associated with COVID-19 complications in the literature are applicable in this context is key for the healthcare system.

We aimed to evaluate the association between sociodemographic, clinical factors and the risk of COVID-19 complications among adults from a community-based healthcare network in Chile. We evaluated the association between sociodemographic, clinical factors and the risk of hospital admissions, admissions to ICU and death due to a COVID-19 infection and estimated the effect of sex and COVID-19 infection in these associations.

## Materials and methods

### Design and data sources

This is a retrospective population-based cohort study using data from electronic health records from all people enrolled in three Primary Care Practices from the Ancora Network, linked to hospital records from the Servicio de Salud Metropolitano Sur-Oriente (SSMSO), the national immunization registry, the mortality and birth records from the Department for National Health Statistics (Departamento de Estadisticas e Información en Salud—DEIS), and records from the National COVID-19 PCR surveillance data. The national COVID-19 PCR surveillance data contained information on all COVID-19 confirmed cases in Chile during the study period, with included PCR and Antigen Rapid Test confirmed cases. The data from all these sources was linked by the SSMSO Statistical Unit using the National Identification Number (RUN) that uniquely identify all citizens in the country, and then anonymized and made available for researchers. We used this dataset as it is the only dataset in Chile that currently links primary healthcare records with hospital records and national registries such as the national immunization registry and mortality records.

The Ancora Network is a local healthcare service network located in the South-East area of the Metropolitan Region in Chile and includes three primary healthcare centres providing care for more than 60,000 people from three different boroughs. The population enrolled in these healthcare centres is younger, with a smaller proportion of people over 65 years old than the national and regional average. However, has a similar distribution in terms of type of public health insurance (S1 Table and S1 Fig) [25, 26].

The Ancora Network electronic health records used in this study include information on patients´ demographic characteristics, recorded diagnoses coded based on the International Classification of Primary Care (ICPC-2), encounters, and prescriptions data.

The data was accessed for research purposes between 21st November 2022 and 31st July 2023. The data was previously anonymized by the data controller and authors did not have access to information that could identify individual participants during or after data analysis.

The aim of this study was to examine risk in the general population rather than in a population infected with COVID-19. This approach has been taken by other researchers to avoid misclassification bias [27], as testing for COVID-19 cases has not always been carried out at community level. Therefore, all patients were included irrespective of any COVID-19 test results.

### Population

We included all people 18 years or older, enrolled in one of the three primary healthcare centers from the Ancora Network between 1st January 2020 and 31st December 2021. People were

excluded if they had an invalid National Identification Number (8 people with invalid RUN), or when no date for enrolment was available (n = 301) due to errors in the dataset.

## Outcome

We considered three outcomes for this study: (1) a hospital admission due to a COVID-19 infection, (2) an ICU admission due to a COVID-19 infection, and (3) death due to a COVID-19 infection that occurred between the 1st January 2020 and the 31st December 2021.

Hospital admissions were identified from the SSMSO hospital records that include all admissions in public and private institutions for people enrolled in any SSMSO primary healthcare centre. We used the date of admission and discharge diagnoses to identify people with hospital admissions due to a COVID-19 infection, and the ward where the patient stayed to identify those who were admitted to the ICU. ICU admissions then include direct admissions to ICU and admissions to ICU from a general ward. COVID-19 cases were identified among those admissions based on PCR test results recorded in the hospital records or the National Covid-19 PCR surveillance data. We only considered admissions due to a COVID-19 infection and only the first admission to hospital for the outcome as only a small proportion (<1%) of the cohort had more than one hospital admission. Duplicates by date of admission were removed. We used death certificate records from DEIS to identify people who died with a diagnosis of COVID-19 as an underlying or contributing cause of death between the 1st January 2020 and the 31st December 2021.

Based on Ayala et al [28], we identified the Pandemic waves were hospital admissions and deaths due to COVID-19 occurred.

## Explanatory variables

**Sociodemographic characteristics.** We identify the date of enrolment, the primary healthcare centre of enrolment, sex and the type of public (FONASA) healthcare insurance of each participant in primary care records from the Ancora Network. People with public health insurance in Chile are classified according to their income in four categories been A the lowest and D the highest income category [26]. We used the public health insurance categories available in primary care records. We calculated the age on the 1st January 2020 from the date of birth available in primary care records.

**Clinical characteristics.** We looked at 14 chronic conditions reported in the literature as associated with COVID-19 complications [12]. As diagnoses and chronic conditions are not systematically coded in the Ancora Network clinical records, we designed a code to identify these 14 conditions in non-structured records of diagnosis available from primary care clinical records data. We used all consultation and prescription records available for each person in the cohort to identify whether they had a record of any of these 14 conditions the year before the date of the outcome or censoring. We extracted all prescriptions filled during the 12-months preceding the outcome of interest or censoring. We identified all prescriptions for chronic conditions and the diagnosis assigned to each prescription. We defined then people with a diagnosis of a chronic condition according to the presence of a) the diagnosis, or b) a prescription generated for that diagnosis, during the year before the outcome or censoring. We counted the number of different chronic conditions identified in each person as a measure of comorbidities as used in previous research [29]. After removing non-prescription drugs (delivery of syringes, food, or creams, etc), we identified repeated prescriptions as those prescription codes dispatched more than three times during the year before the outcome. We counted the number of repeated prescriptions during this period as a measure of comorbidities as used in previous research [29]. All codes used are available in the S2 Table.

Using data from the electronic health record we calculated the number of encounters (virtual and face-to-face appointments) that people had with a physician or a nurse in the primary healthcare centre the year before the outcome or censoring (12 months) using the date, type of contact and healthcare professional involved.

The National Immunisation registry contains information on all vaccine doses administered in Chile. We identified all people who had an influenza vaccine, and the number of COVID-19 vaccine doses the year before the outcome or censoring from National Immunisation records. National birth records from the DEIS contain information on date of birth, gestational week and information about the mother of all birth registered in the country. Information was linked to the SSMSO data using the National Identification Number of the mother recorded in birth certificates. This information was used to identify women who were pregnant or puerperal before the outcome or censoring. We use the National COVID-19 PCR surveillance data to identify people with a positive result of a COVID-19 PCR test during the study period.

**Analysis.** Data were described using count and percentage for categorical variables and mean and SD for continuous variables. Chi2 and t-test were used to analysed associations between the outcome variables and categorical or continuous variables respectively. We plotted the distribution of hospital admissions due to COVID-19 infection, admissions to ICU due to COVID-19 infection and deaths due to COVID-19 infection between the 1st January 2020 and the 31st December 2021 observed for the entire cohort by epidemiological week.

We calculated the follow-up time based on the date the participant was registered in the healthcare centre or the study starting date (1st January 2020) whichever was later, and the date of the event (hospital admission or death) or the end of the study period (31st December 2021) whichever was first.

We used a Kaplan-Meier failure function to explore the relationship between each explanatory variables and the three outcomes of the study. Results from this exploration are available in S2 Fig. We used Cox proportional hazard model to evaluate the association between the explanatory variables and each of the outcomes of the study separately. Multivariate Cox proportional hazard models were fitted using the stset command in Stata, specifying the healthcare primary care centre as a strata variable. Age, sex, type of FONASA insurance, number of encounters with physicians and nurses in primary care, influenza and COVID-19 vaccine doses were forced in the model. Age was categorised to understand the effect in people older than 70 years. For the outcome deaths due to COVID-19 infection, we used as reference the category 18 to 54 year as no person younger than 35 died from COVID-19 during this period [30]. As the count of comorbidities and frequently dispatched drugs are correlated, we only included the count of dispatched drugs as some of the comorbidities identified were underreported in primary care records [30]. This approach to assessing multimorbidity performs similarly to other multimorbidity indexes such as the Charlson Comorbidity Index or the count of comorbidities to predict 3-year mortality [29]. The number of frequently dispatched drugs was categorised in Table 1 to facilitate comprehension, but included as a continuous variable in the model. We also included in the models the conditions hypertension, diabetes and depression to explore their independent effect in the outcomes. We did not include other chronic conditions as their proportion in the cohort was significantly lower than expected based on estimations from the National Health Survey and we hypothesise they were likely being underreported in the data. Missing values in explanatory variables was treated as missing at random and excluded from the model as the proportion was small (<5%).

We tested the proportionality assumption for explanatory variables in the model using the Test of proportional hazards assumption (phtest) on the basis of Schoenfeld residuals after fitting the model. Results from this analysis are in S3 Table. As some of the explanatory variables

**Table 1. Sociodemographic and clinical characteristics of participants by outcome of the study.**

| | Total | | COVID positive test | | A hospital admission due to COVID-19 infection | | | | | An ICU admission due to COVID-19 infection | | | | | Death due to COVID-19 infection | | | | |
|---|---|---|---|---|---|---|---|---|---|---|---|---|---|---|---|---|---|---|---|
| | | | | | No | | Yes | | | No | | Yes | | | No | | Yes | | |
| | n = 44,674 | | n = 6185 | | n = 44,219 | | n = 455 | | | n = 44,458 | | n = 216 | | | n = 44,526 | | n = 148 | | |
| | No. | % | No. | % | No. | % | No. | % | p-value | No. | % | No. | % | p-value | No. | % | No. | % | p-value |
| Days of follow-up | | | | | | | | | | | | | | | | | | | |
| Mean (sd) | 712.90 | (84.27) | 711.20 | (86.10) | 713.67 | (82.29) | 637.52 | (182.75) | <0.001 | 713.34 | (83.10) | 622.15 | (197.94) | <0.001 | 714.20 | (80.79) | 320.91 | (160.49) | <0.001 |
| Age | | | | | | | | | | | | | | | | | | | |
| Mean (sd) | 44.30 | (17.31) | 44.02 | (16.79) | 44.15 | (17.27) | 58.39 | (15.70) | <0.001 | 44.24 | (17.31) | 55.77 | (14.49) | <0.001 | 44.22 | (17.27) | 68.54 | (11.66) | <0.001 |
| Age categories (Ref 18 to 34) | | | | | | | | | | | | | | | | | | | |
| 35 to 54 | 15,513 | 34.7 | 2257 | 36.50 | 15,387 | 34.80 | 126 | 27.70 | <0.001 | 15,445 | 34.70 | 68 | 31.50 | <0.001 | 15,495 | 34.80 | 18 | 12.20 | <0.001 |
| 55 to 69 | 9157 | 20.5 | 1258 | 20.30 | 8993 | 20.30 | 164 | 36.00 | | 9068 | 20.40 | 89 | 41.20 | | 9100 | 20.40 | 57 | 38.50 | |
| >70 | 3782 | 8.5 | 455 | 7.40 | 3664 | 8.30 | 118 | 25.90 | | 3746 | 8.40 | 36 | 16.70 | | 3709 | 8.30 | 73 | 49.30 | |
| Sex (Ref Male) | | | | | | | | | | | | | | | | | | | |
| Female | 24,855 | 55.6 | 3552 | 57.40 | 24,636 | 55.70 | 219 | 48.10 | 0.001 | 24,762 | 55.70 | 93 | 43.10 | <0.001 | 24,788 | 55.70 | 67 | 45.30 | 0.011 |
| Tramo Fonasa (Ref A lowest income) | | | | | | | | | | | | | | | | | | | |
| B | 15,226 | 34.1 | 2060 | 33.30 | 15,030 | 34.00 | 196 | 43.10 | <0.001 | 15,158 | 34.10 | 68 | 31.50 | 0.723 | 15,149 | 34.00 | 77 | 52.00 | <0.001 |
| C | 7514 | 16.8 | 1145 | 18.50 | 7440 | 16.80 | 74 | 16.30 | | 7474 | 16.80 | 40 | 18.50 | | 7494 | 16.80 | 20 | 13.50 | |
| D (highest income) | 12,741 | 28.5 | 1835 | 29.70 | 12,630 | 28.60 | 111 | 24.40 | | 12,675 | 28.50 | 66 | 30.60 | | 12,713 | 28.60 | 28 | 18.90 | |
| Missing | 2082 | 4.7 | 142 | 2.30 | 2074 | 4.70 | 8 | 1.80 | | 2075 | 4.70 | 7 | 3.20 | | 2079 | 4.70 | 3 | 2.00 | |
| Count of comorbidities (Ref 0) | | | | | | | | | | | | | | | | | | | |
| 1 to 2 | 11,539 | 25.8 | 2133 | 34.50 | 11,327 | 25.60 | 212 | 46.60 | <0.001 | 11,437 | 25.70 | 102 | 47.20 | <0.001 | 11,465 | 25.70 | 74 | 50.00 | <0.001 |
| > = 3 | 1050 | 2.4 | 214 | 3.50 | 1011 | 2.30 | 39 | 8.60 | | 1035 | 2.30 | 15 | 6.90 | | 1029 | 2.30 | 21 | 14.20 | |
| Frequently dispatched drugs (Ref 0) | 32,730 | 73.3 | 3906 | 63.20 | 32,495 | 73.50 | 235 | 51.60 | <0.001 | 32,615 | 73.40 | 115 | 53.20 | <0.001 | 32,643 | 73.30 | 87 | 58.80 | <0.001 |
| 1 to 2 | 3403 | 7.6 | 532 | 8.60 | 3376 | 7.60 | 27 | 5.90 | <0.001 | 3391 | 7.60 | 12 | 5.60 | | 3399 | 7.60 | 4 | 2.70 | |
| 3 to 5 | 4317 | 9.7 | 800 | 12.90 | 4246 | 9.60 | 71 | 15.60 | | 4285 | 9.60 | 32 | 14.80 | | 4304 | 9.70 | 13 | 8.80 | |
| > = 6 | 4224 | 9.5 | 947 | 15.30 | 4102 | 9.30 | 122 | 26.80 | | 4167 | 9.40 | 57 | 26.40 | | 4180 | 9.40 | 44 | 29.70 | |
| Record of Comorbidities in PC | | | | | | | | | | | | | | | | | | | |
| HTA (Ref No) | 5198 | 11.6 | 917 | 14.80 | 5073 | 11.50 | 125 | 27.50 | <0.001 | 5142 | 11.60 | 56 | 25.90 | <0.001 | 5149 | 11.60 | 49 | 33.10 | <0.001 |
| DM (Ref No) | 4181 | 9.4 | 782 | 12.60 | 4067 | 9.20 | 114 | 25.10 | <0.001 | 4132 | 9.30 | 49 | 22.70 | <0.001 | 4131 | 9.30 | 50 | 33.80 | <0.001 |
| Depression (Ref No) | 3506 | 7.8 | 769 | 12.40 | 3458 | 7.80 | 48 | 10.50 | 0.031 | 3480 | 7.80 | 26 | 12.00 | 0.022 | 3488 | 7.80 | 18 | 12.20 | 0.051 |
| Mean (sd) number of encounters with | | | | | | | | | | | | | | | | | | | |
| Physicians | 0.02 | (0.16) | 0.03 | (0.19) | 0.02 | (0.16) | 0.04 | (0.21) | 0.001 | 0.02 | (0.16) | 0.05 | (0.24) | 0.002 | 0.02 | (0.16) | 0.04 | (0.23) | 0.081 |
| Nurses | 0.17 | (0.75) | 0.33 | (0.99) | 0.16 | (0.74) | 0.53 | (1.40) | <0.001 | 0.16 | (0.75) | 0.42 | (0.89) | <0.001 | 0.16 | (0.74) | 0.87 | (2.46) | <0.001 |
| Influenza vaccine previous year (Ref No) | 3421 | 7.7 | 591 | 9.60 | 3364 | 7.60 | 57 | 12.50 | <0.001 | 3397 | 7.60 | 24 | 11.10 | 0.056 | 3410 | 7.70 | 11 | 7.40 | 0.918 |
| Number of COVID vaccine doses (Ref 0) | | | | | | | | | | | | | | | | | | | |
| 1 | 1695 | 3.8 | 238 | 3.80 | 1657 | 3.70 | 38 | 8.40 | <0.001 | 1677 | 3.80 | 18 | 8.30 | <0.001 | 1688 | 3.80 | 7 | 4.70 | <0.001 |
| 2 | 7702 | 17.2 | 1259 | 20.40 | 7613 | 17.20 | 89 | 19.60 | | 7665 | 17.20 | 37 | 17.10 | | 7677 | 17.20 | 25 | 16.90 | |
| 3 | 29,695 | 66.5 | 4037 | 65.30 | 29,577 | 66.90 | 118 | 25.90 | | 29,646 | 66.70 | 49 | 22.70 | | 29,695 | 66.70 | 0 | 0.00 | |
| Pregnant or puerperal women (Ref No) | 481 | 1.1 | 115 | 1.90 | 471 | 1.10 | 10 | 2.20 | 0.020 | 477 | 1.10 | 4 | 1.90 | 0.268 | 481 | 1.10 | 0 | 0.00 | 0.204 |
| Covid positive (Ref No) | 6185 | 13.8 | 6185 | 100 | 5730 | 13.00 | 455 | 100 | <0.001 | 5969 | 13.40 | 216 | 100 | <0.001 | 6037 | 13.60 | 148 | 100 | <0.001 |
| Covid hospital admission (Ref No) | 455 | 1.0 | 455 | 7.40 | - | | - | | <0.001 | 239 | 0.50 | 216 | 100 | <0.001 | 368 | 0.80 | 87 | 58.80 | <0.001 |
| Covid-ICU hospital admission (Ref No) | 216 | 0.5 | 216 | 3.50 | 0 | 0.00 | 216 | 47.50 | <0.001 | - | | - | | | 171 | 0.40 | 45 | 30.40 | <0.001 |
| Covid death (Ref No) | 148 | 0.3 | 148 | 2.40 | 61 | 0.10 | 87 | 19.10 | <0.001 | 103 | 0.20 | 45 | 20.80 | <0.001 | - | | - | | |

(age, sex and COVID-19 vaccine doses) violated of the proportional assumption in some of the models, we fitted models specifying those variables to vary continuously with respect to time, using the interaction with the logarithm of analysis time.

We used two subgroup analyses. The first subgroup analysis was used to explore the effect of sex and pregnancy on the outcomes of this study. As no pregnant or puerperal women died of COVID-19 in this cohort, pregnancy was excluded from the models for COVID-19 death. The second subgroup analysis was used to explore the risk of having a COVID-19 complication only among people who had a confirmed COVID-19 infection during the study period

We used Stata© version 17.0 MP-parallel Edition for the analysis.

### Ethical statement

This study was approved by the Pontificia Universidad Católica de Chile Etical Committee (ID 220570006) in July 2022 and by the Servicio de Salud Metropolitano Sur Oriente Etical Committee in October 2022.

## Results

We identified 44,983 people older than 18 years enrolled in one of the three Ancora Network Primary care centres between the 1st January 2020 and 31st December 2021. In 301 of them, there was no date of enrolment and in eight of them, the National Identification number was invalid and therefore, were excluded. The 44,674 people included were distributed as 12,192 in centre 1, 15,996 in centre 2 and 16,486 in centre 3 (Fig 1). Sociodemographic characteristics of the cohort by centre can be found in S4 Table.

### Sociodemographic and clinical characteristics

The mean age of the cohort was 44.3 (sd 17.31) years old, with 36.3% of the cohort being younger than 34 years old. 55.6% of the cohort was female, 15.9% had a type of healthcare insurance for people from the lowest category of income and 28.5% for people from the highest category of income. The majority of the cohort (71.8%) had no chronic comorbidities, 11.6% and 9.4% had a record of hypertension or diabetes mellitus diagnosis in their clinical records respectively, and 19.2% had three or more frequently dispatched drugs in the past year. We found 481 women who were pregnant or puerperal the year before the outcome or the end of the study. 87.5% of the cohort had at least one COVID-19 vaccine dose during the period of study and only 7.7% had an influenza vaccine the year before. Only 6185 people had a positive PCR for COVID-19 infection between 2020 and 2021, and 1.2% had more than one (Table 1). The mean follow-up period for the entire cohort was 712.9 (SD 84.27) days.

### Hospital admissions and deaths due to COVID-19 infection

Among the 44,674 people in the cohort, 455 (1.02%) had a hospital admission due to a COVID-19 infection, and 216 of them were also admitted to the ICU due to COVID-19 infection between 2020 and 2021. Of the 455 hospital admissions identified in the cohort, 161 (35.38%) occurred during the first Pandemic wave, 221(48.57%) during the second Pandemic wave, 5(1.1%) during the third Pandemic wave, and 68(14.95%) occurred between wave periods (Fig 2).

We identified 533 decedents in the cohort between 2020 and 2021, 148 of them died with COVID-19 as an underlying or contributing cause of death. Of the 455 people admitted to hospital due to COVID-19 infection, 87 also died with COVID-19 during the study period. Of the 148 COVID-19 deaths identified in the cohort, 68(45.95%) occurred during the first Pandemic

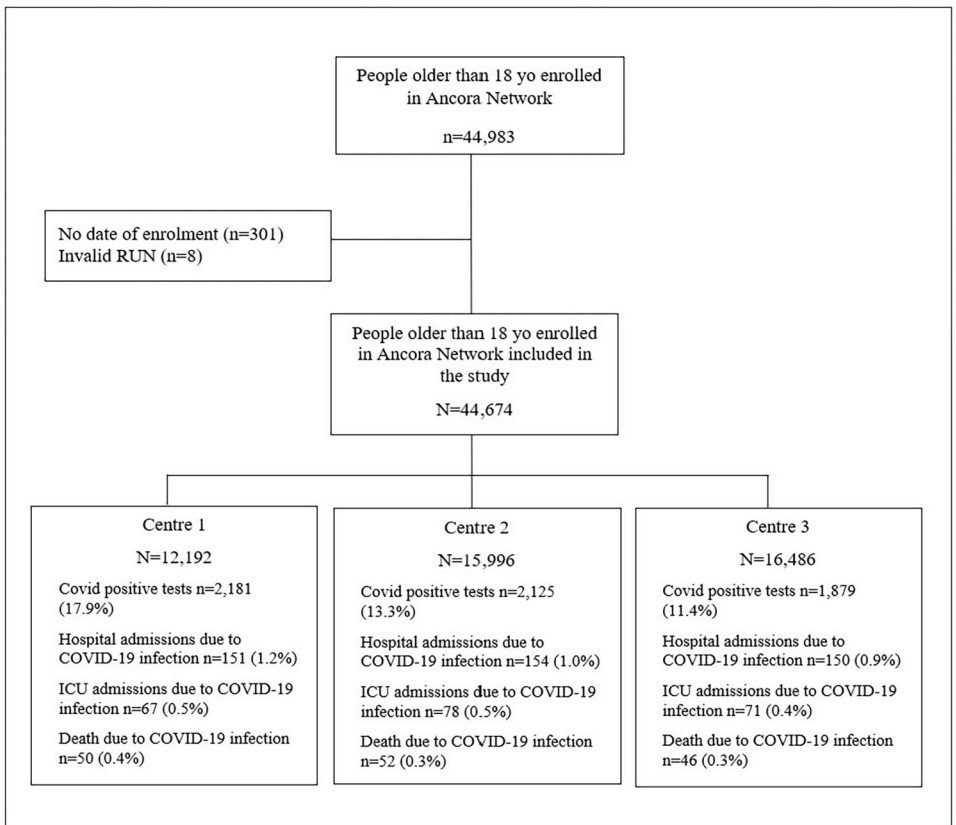

**Fig 1. Flow diagram of the cohort.** The diagram shows the numbers of people (n) excluded at different stages and the final number of people included and with an identification of cases for the outcomes by centre.

wave, 38(25.68%) during the second Pandemic, and 42(28.38%) occurred between wave periods.

Fig 2 shows the distribution of hospital admissions due to COVID-19 infection, ICU admissions due to COVID-19 infection and deaths due to COVID-19 infection in the cohort by epidemiological week in 2020 and 2021.

People with a hospital admission due to COVID-19 infection, an ICU admission due to COVID-19 infection or death due to COVID-19 infection were older, had a higher count of comorbidities and frequently dispatched drugs, had a higher prevalence of hypertension, diabetes and depression, had more contacts with physicians and nurses during the previous year and were more likely to have no COVID-19 vaccine doses (Table 1).

## Multivariate Cox proportional hazard model

**Sociodemographic factors.**　Age was consistently associated with the three outcomes of the study and interacted with time for hospital and ICU admissions. People 70 years or older had 1.53 times (HR 1.53; 95% CI 1.43 to 1.64) and 1.40 (HR 1.40; 95% CI 1.26 to 1.55) higher risk of being admitted to hospital or have an ICU admission due to a COVID-19 infection than those between 18 and 34 years old respectively during the study period. People 70 years or older had 1.88 times (HR 1.88; 95% CI 1.69 to 2.08) higher risk of dying due to a COVID-19 infection during the period of study than those aged 54 years or younger (Table 2 and Fig 3).

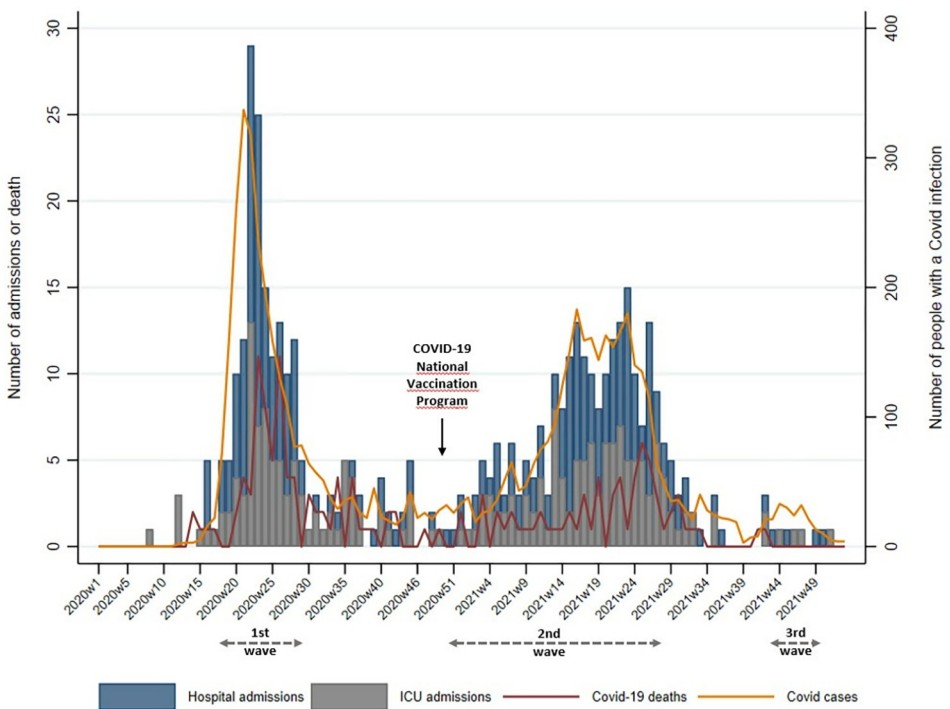

**Fig 2. Distribution of hospital admissions due to COVID-19 infection, ICU admissions due to COVID-19 infection and deaths due to COVID-19 infection in the population cohort by epidemiological week.**

Compared with male sex, females were 0.96 (95% CI 0.93 to 1.00), 0.94 (95% CI 0.89 to 0.99) and 0.64 (95% CI 0.45 to 0.91) times less likely to be admitted to hospital, admitted to ICU or die due to a COVID-19 infection (Table 2 and Fig 3). The type of healthcare insurance was associated with the risk of a hospital admission due to a COVID-19 infection but not with an ICU admission or death due to COVID-19 infection. People with healthcare insurance from the highest category of income (FONASA D) were 1.38 times (95% CI 1.01 to 1.88) more likely to be admitted to hospital due to a COVID-19 infection than those from the lowest category of income (FONASA A) (Table 2).

## Clinical factors

People with a diagnosis of hypertension and diabetes mellitus were more likely to be admitted to hospital and die due to a COVID infection, but these two comorbidities were not associated to ICU admissions due to COVID-19 infection. The number of frequently dispatched drugs was associated with a higher risk of a hospital (HR 1.10; 95% CI 1.07 to 1.14) and ICU admission (HR 1.13; 95% CI 1.08 to 1.17) due to COVID-19 infection but not with death due to COVID-19 infection (HR 1.03; 95% CI 0.98 to 1.08).

The number of COVID vaccine doses was consistently associated with a lower risk of hospital, ICU admissions, and death due to a COVID-19 infection, with a clear gradient in the reduction of the risk by number of doses. People with three or more doses of COVID vaccine had a 0.57 lower risk (HR 0.57; 95% CI 0.54 to 0.59) of being admitted to hospital due to COVID-19 infection than those with no dose. People with two or more COVID vaccine doses had 0.60 (HR 0.60; 95% CI 0.57 to 0.63) and 0.50 (HR 0.50; 95% CI 0.46 to 0.54) the risk of having an ICU admission or dying due to COVID-19 infection than those without a COVID vaccine respectively (Table 2 and Fig 3).

**Table 2. Hazard Ratio of the risk for a hospital admission, an ICU admission or death due to COVID-19 infection between 2020 and 2021.**

| | A hospitalization due to COVID-19 infection | | | An UCI/UTI hospital admission due to COVID-19 infection | | | Death due to COVID-19 infection | | |
|---|---|---|---|---|---|---|---|---|---|
| | n = 42,592 | | | n = 42,592 | | | n = 42,592 | | |
| | HR | 95% CI | p-value | HR | 95% CI | p-value | HR | 95% CI | p-value |
| Age categories (Ref 18 to 34) | | | | | | | | | |
| 35 to 54 | 1.27 | (1.20–1.35) | <0.001 | 1.26 | (1.16–1.37) | <0.001 | | | |
| 55 to 69 | 1.46 | (1.38–1.56) | <0.001 | 1.44 | (1.32–1.57) | <0.001 | 1.60* | (1.45–1.76) | <0.001 |
| >70 | 1.53 | (1.43–1.64) | <0.001 | 1.40 | (1.26–1.55) | <0.001 | 1.88* | (1.69–2.08) | <0.001 |
| Sex (Ref Male) | | | | | | | | | |
| Female | 0.96 | (0.93–1.00) | 0.035 | 0.94 | (0.89–0.99) | 0.012 | 0.64 | (0.45–0.91) | 0.012 |
| Tramo fonasa (Ref A lowest income) | | | | | | | | | |
| B | 1.20 | (0.89–1.62) | 0.220 | 0.90 | (0.58–1.38) | 0.620 | 1.00 | (0.59–1.69) | 0.992 |
| C | 1.45 | (1.03–2.03) | 0.032 | 1.46 | (0.92–2.32) | 0.109 | 1.02 | (0.54–1.94) | 0.953 |
| D (highest income) | 1.38 | (1.01–1.88) | 0.042 | 1.44 | (0.94–2.18) | 0.092 | 1.07 | (0.59–1.91) | 0.832 |
| Record of Comorbidities in Primary Care | | | | | | | | | |
| HTA (Ref No) | 1.39 | (1.09–1.76) | 0.007 | 1.37 | (0.96–1.95) | 0.080 | 1.42 | (0.97–2.09) | 0.072 |
| DM (Ref No) | 1.39 | (1.08–1.80) | 0.011 | 1.23 | (0.83–1.81) | 0.299 | 2.04 | (1.36–3.06) | 0.001 |
| Depression (Ref No) | 0.97 | (0.70–1.33) | 0.832 | 1.24 | (0.79–1.93) | 0.350 | 1.46 | (0.86–2.47) | 0.165 |
| Frequently dispatched drugs | 1.10 | (1.07–1.14) | <0.001 | 1.13 | (1.08–1.17) | <0.001 | 1.03 | (0.98–1.08) | 0.219 |
| Number of encounters with Physicians | 1.42 | (0.97–2.07) | 0.068 | 1.68 | (1.02–2.75) | 0.040 | 1.56 | (0.79–3.05) | 0.199 |
| Number of encounters with nurses | 1.04 | (0.98–1.10) | 0.188 | 1.02 | (0.92–1.14) | 0.663 | 1.13 | (1.06–1.22) | 0.001 |
| Influenza vaccine previous year (Ref No) | 1.09 | (1.03–1.15) | 0.003 | 1.23 | (0.77–1.96) | 0.378 | 0.97 | (0.86–1.09) | 0.575 |
| Number of Covid vaccine doses received (Ref 0) | | | | | | | | | |
| 1 | 0.97 | (0.92–1.03) | 0.350 | 0.94 | (0.86–1.02) | 0.135 | 0.86** | (0.76–0.98) | 0.028 |
| 2 | 0.8 | (0.77–0.84) | <0.001 | 0.60 | (0.57–0.63) | <0.001 | 0.50** | (0.46–0.54) | <0.001 |

(*Continued*)

**Table 2.** (Continued)

| | A hospitalization due to COVID-19 infection | | | An UCI/UTI hospital admission due to COVID-19 infection | | | Death due to COVID-19 infection | | |
|---|---|---|---|---|---|---|---|---|---|
| | n = 42,592 | | | n = 42,592 | | | n = 42,592 | | |
| 3 | 0.57 | (0.54–0.59) | <0.001 | | | | | | |

HTA: hypertension; DM: diabetes mellitus

(*) Reference 18 to 55 yo

(**) Reference 0, categories 1, 2+ doses.

[1]: age, sex, influenza vaccine and COVID-19 vaccines included as time-varying covariate

[2]: age, sex and COVID-19 vaccines included as time-varying covariate

[3]: age, influenza vaccine and COVID-19 vaccines included as time-varying covariate.

## Subgroup analysis

The first subgroup analysis by sex shows that the increase in the risk of having a hospital admission, an ICU admission or death due to a COVID-19 infection by age was higher for males than females. The Hazard Ratio of a hospital admission and ICU admission due to COVID-19 infection for males older than 70 years (versus those younger than 34 years) was 1.8 (95% CI 1.60 to 2.03) and 1.68 (95% CI 1.43 to 1.99) in contrast to 1.37(95% CI 1.26 to 1.50) and 1.17 (95% CI 1.01 to 1.37) for females respectively (Fig 4 and S5 Table). A similar pattern is observed for deaths due to COVID-19 infection (Fig 4 and S5 Table).

The risk of hospital admission due to COVID-19 infection and deaths due to COVID-19 infection for people with hypertension and diabetes mellitus also differed by sex. A diagnosis

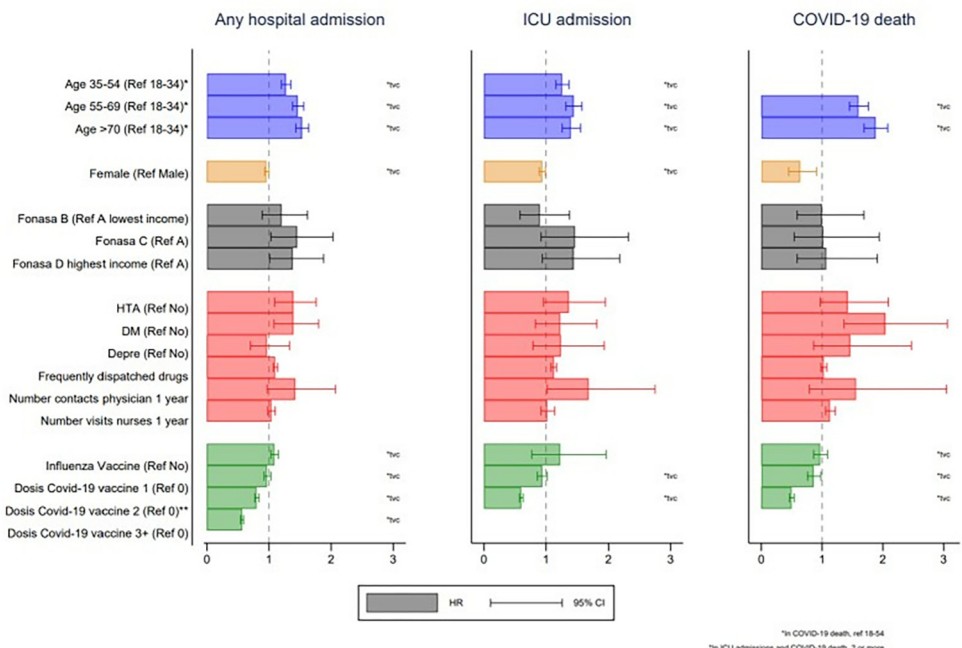

**Fig 3. Hazard Ratio of the risk for a hospital admission, ICU admission or death due to COVID-19 infection between 2020 and 2021.**

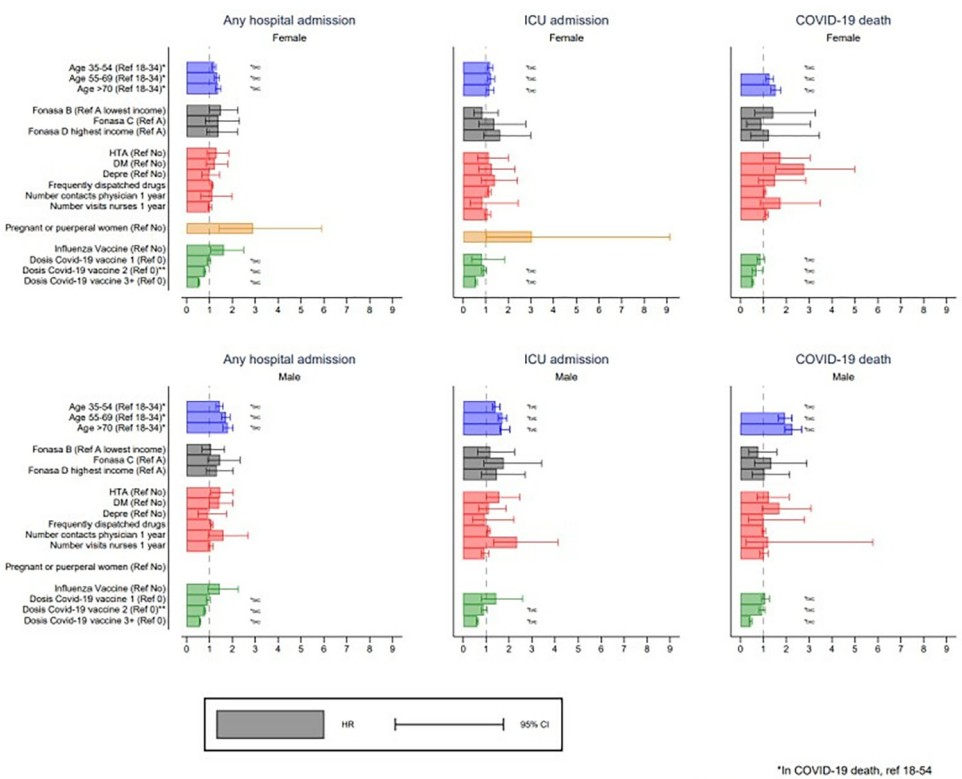

**Fig 4. Subgroup analysis: Hazard Ratio of the risk for a hospital admission, ICU admission or death due to COVID-19 infection between 2020 and 2021, by sex.**

of hypertension was associated with a higher risk of hospital admission due to COVID-19 infection among males (HR 1.47; 95% CI 1.06 to 2.04) but not among females (HR 1.31; 95% CI 0.91 to 1.84). However, the diagnosis of hypertension was associated with a higher risk of deaths due to COVID-19 infection among females (HR 1.73; 95% CI 0.99 to 3.04) but not among males (HR 1.24; 95% CI 0.72 to 2.12) (Fig 4 and S5 Table).

Female that were pregnant or puerperal during the period of exposure were 2.89 (95% CI 1.41 to 5.89) and 3.04 (95% CI 1.01 to 9.14) times more likely to have a hospital admission or ICU admission due to COVID-19 infection than those who were not pregnant of puerperal respectively (Fig 4 and S5 Table). No pregnant women in this cohort died due to a COVID infection.

The second subgroup analysis among people with a confirmed COVID-19 infection shows similar results, with Hazard Ratio for age and COVID-19 vaccine doses higher than the Hazard Ratio in the models for the entire cohort. Sex is no longer statistically significantly associated with any of the three outcomes when including only people with a confirmed COVID-19 infection (S6 Table).

## Discussion

In this population-based study using a cohort of people enrolled in a large Primary Care Network in Chile, we identify a prevalence of COVID-19 complications of 1.02% for hospital admission, 0.48% for ICU admission and 0.33% for deaths due to COVID-19 infection. Older age and male sex were associated with a higher risk of these three COVID-19 complications.

Hypertension and diabetes mellitus conditions were associated with a higher risk of a hospital admission and death due to COVID-19 infection, but not with ICU admissions and being pregnant was associated with a higher risk of a hospital and ICU admission due to COVID-19 infection. Two or more COVID-19 vaccine doses was consistently associated with a lower risk of these three outcomes.

To our knowledge, this is the first population-based study exploring factors associated with COVID-19 complications in Chile. Most of the evidence in risk factors for COVID-19 complications comes from Europe, USA and Asia, which might not necessarily be useful for the Latin American context. In Treskova-Schwarzbach et al [12], 160 primary studies from 120 systematic reviews investigating risk factors for COVID-19 complications were retrieve, and only 10 were from a Latin-American country: six from Mexico [31–35], three from Brazil [36–38], Five articles exploring factors associated with COVID-19 complications or severity in Chile have been published since then [39–43]. However, all these studies analysed risk factors for hospital patients and were not population-based, which might overestimate the risk at a population level.

We found older age and male sex were consistently and significantly associated with a higher risk of the three COVID-19 complications analysed in this study. These findings are consistent with findings from other regions [8, 44]. Furthermore, older age was the strongest predictor of COVID-19 death, after adjusting by comorbidities and COVID-19 vaccination. Different effects of age in COVID-19 outcomes have been described in the literature. In Starke et al systematic review, age was lineally associated with a higher risk of COVID-19 mortality after adjusting for comorbidities, with no evidence of a specific age threshold and no significant association with ICU admissions [45]. Instead, in Bonanad et al systematic review, a larger increase in mortality risk was observed in patients aged 60 to 69 years compared with those aged 50 to 59 years, and an exponential increase in mortality risk by age was observed [46]. In our study, age shows a significant trend towards a higher COVID-19 mortality risk, with a higher HR in older adults. However, a different pattern is observed for hospital and ICU admissions. People aged 70 years and older had a lower HR for hospital and ICU admissions due to COVID-19 infection comparing to those aged between 18 and 34, than those between 55 and 69 years old. This is likely to be the effect of hospital protocols to prioritise scares healthcare resources towards younger adults with higher chances of survival, rather than a biological effect of COVID-19.

We found people with hypertension and diabetes mellitus had a higher risk of a hospital admission and death due to COVID-19 infection. These findings are consistent with the international literature. Treskova-Schwarzbach et al systematic review, results from the meta-analysis of high-quality studies, the risk of hospital admission for people with diabetes mellitus ranged from 1.77 to 2.03, and the risk of COVID-death ranged from 1.21 to 2.02 [12], similar to the 1.36 and 2.04 HR found in this study. Hypertension has also been associated with a higher risk of hospital admission in the European region and COVID-19 mortality, with Odds Ratios between 1.3 and 1.69 [12]. While people with diabetes and hypertension in our cohort had a higher risk of ICU admission, this association was not statistically significant unlike results from international studies. In Booth et al systematic review, hypertension was associated with a higher risk of COVID-19 severity and mortality in the univariate analysis, but the risk of COVID-19 severe presentation was not statistically significant in the multivariate pooled risk analysis [8]. Instead, obesity showed a strong correlation with COVID-19 severity and mortality in both univariate and multivariate pooled risk analysis. In an observational study in Mexican population, Bello-Chavolla et al found that obesity mediate 49.5% of the effect of diabetes on COVID-19 mortality and that conferred an increased risk for ICU admission [35]. We were not able to identify people with obesity or overweight in our cohort as it

was not available in the dataset, and therefore we could not account for this important confounder.

We found women who were pregnant or puerperal during the study period had 2.89- and 3.04-times higher chances of being admitted to hospital or ICU due to COVID-19 infection respectively, but no pregnant women in our cohort died from COVID-19. Two systematic reviews based on case reports have reported higher rate of adverse pregnancy outcomes, including preterm birth and pregnancy loss, as well as severe morbidity in pregnant women with COVID-19 infection, but clinical presentations similar to the general population [13, 14]. In Khan et al systematic review, pregnant women with a COVID-19 infection were more likely to be asymptomatic than non-pregnant women. The pooled relative risk from the meta-analysis of more than 400 thousand participants found no difference in the risk of COVID-19 severity, but a higher risk of ICU admission amongst pregnant women [15]. In an analysis of hospital admissions among pregnant women in eight United State hospitals during the COVID-19 pandemic, the most frequent reason for admission were obstetric reasons [47]. In this study, we included all hospital admissions associated to a COVID-19 infection. The higher risk of hospital admission among pregnant women with COVID-19 infection might be explained by pregnant women who were admitted for other non-COVID-19 related reasons and that were infected in hospital.

We found a strong association between COVID-19 vaccination and a lower risk of COVID-19 complications, with a significant and decreasing gradient in the risk of all COVID-19 complications by COVID-19 number of doses. Safety and efficacy of COVID-19 vaccines has been widely reported [48–50]. Chile initiated the COVID-19 vaccine programme in December 2020. By January 2022, 44 million COVID-19 vaccine doses had been administrated and 86% of the population had received at least one COVID-19 vaccine dose [51]. Accounting for the number of COVID-19 vaccine doses people received is an important strength of this study.

## Strengths and limitations

This study has several strengths. It is a population-based study using data from primary care records linked to five national datasets, which allow us to retrieve all hospital admissions, deaths, immunisation, births and COVID-19 PCR results occurred in this population, reducing the risk of selection and information bias. This design allowed us to access a large cohort of exposed people during the period of study, being one of the largest cohort studies exploring risk factors for COVID-19 complications, contributing the knowledge in this field for the Latin American population. Nevertheless, this study has some limitations. The population included is not necessarily representative of the country. Hospital capacity, in particular during COVID-19 waves, might have prevented some people to be admitted to hospital or ICU even when it was clinically appropriate, which might affect risk factor estimators. Nevertheless, hospital and ICU admissions derived to the private sector when public health hospital were loaded were also included. We were not able to include all pre-existing chronic conditions associated with COVID-19 infection in the literature, such as obesity, chronic kidney disease, symptoms, diabetes and hypertension level of control, or factors such as mechanical ventilation or wave of infection due to the lack of structured and systematic recording in primary and hospital healthcare records. We included all people registered in the Ancora Primary Care Network and not just people who had been infected, assuming all the population had the same risk of COVID-19 infection. Lockdown measures in Chile were implemented by borough and therefore, it is possible to assume that most of the population living in the same borough had a similar exposure to COVID-19. However, people´s behaviours might have also determined the

level of exposure and risk to COVID-19 complications. We were not able to assess this in our study. Nevertheless, including only people with a positive COVID-19 PCR or Antigen Rapid Test would have sub estimate the number of people infected during this period, as asymptomatic people were not routinely tested.

## Conclusion

In this population-based study using a cohort of people enrolled in a large Primary Care Network in Chile, sociodemographic and clinical factors such as age, sex and pre-existing comorbidities were significantly associated with COVID-19 complications. Risk factors were comparable to those reported in similar studies from other high-income countries, and can be used to predict severity in COVID-19 patients. COVID-19 vaccination was the most significant protective factor for COVID-19 complications.

## Supporting information

**S1 Fig. Map of Santiago with location of the three primary healthcare centres.**
(TIF)

**S2 Fig. Kaplan-Meier failure functions.**
(DOCX)

**S1 Table. Comparing national, regional and Ancora Network population´s characteristics.**
(DOCX)

**S2 Table. Codes to define comorbidities.**
(DOCX)

**S3 Table. Results from the test of proportional hazards assumption (phtest).**
(DOCX)

**S4 Table. Sociodemographic and clinical characteristics by healthcare centre.**
(DOCX)

**S5 Table. Hazard Ratio for subgroup analysis by sex.**
(DOCX)

**S6 Table. Hazard Ratio for subgroup analysis for people with a confirmed COVID-19 infection.**
(DOCX)

## Author Contributions

**Conceptualization:** Javiera Leniz, Mauricio Soto, Eduardo Arenas, Paula Margozzini, Francisco Suarez, Daniel Capurro, María Paulina Rojas, Claudia Bambs.

**Data curation:** Javiera Leniz, Sam Hernández-Jaña, Mauricio Soto, Eduardo Arenas, Francisco Suarez.

**Formal analysis:** Javiera Leniz, Sam Hernández-Jaña.

**Investigation:** Javiera Leniz, Sam Hernández-Jaña, Paula Margozzini, María Paulina Rojas, Claudia Bambs.

**Methodology:** Javiera Leniz, Mauricio Soto, Paula Margozzini, Daniel Capurro, María Paulina Rojas, Claudia Bambs.

**Project administration:** Javiera Leniz.

**Software:** Javiera Leniz, Sam Hernández-Jaña, Eduardo Arenas, Francisco Suarez.

**Supervision:** Javiera Leniz, Paula Margozzini, María Paulina Rojas, Claudia Bambs.

**Validation:** Javiera Leniz, Mauricio Soto, Eduardo Arenas, Paula Margozzini, Francisco Suarez, Daniel Capurro, María Paulina Rojas, Claudia Bambs.

**Visualization:** Javiera Leniz.

**Writing – original draft:** Javiera Leniz.

**Writing – review & editing:** Javiera Leniz, Sam Hernández-Jaña, Mauricio Soto, Eduardo Arenas, Paula Margozzini, Francisco Suarez, Daniel Capurro, María Paulina Rojas, Claudia Bambs.

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
