## [Decision Letter · Decision Letter 0]

20 May 2024

PONE-D-24-13634Association between demographic, clinical characteristics and severe complications by SARS-CoV-2 infection in a community-based healthcare network in ChilePLOS ONE

Dear Dr. Leniz,

Thank you for submitting your manuscript to PLOS ONE. After careful consideration, we feel that it has merit but does not fully meet PLOS ONE’s publication criteria as it currently stands. Therefore, we invite you to submit a revised version of the manuscript that addresses the points raised during the review process.

Please review and address the comments from the reviewers, in particular the methodological aspects expressed by reviewer 2. Also, please update the references to reflect better what is already know in the topic and what your study is adding. ==============================

We look forward to receiving your revised manuscript.

Kind regards,

Juan Pablo Gutierrez

Academic Editor

PLOS ONE

2. In the online submission form, you indicated that your data is available only on request from a third party. Please note that your Data Availability Statement is currently missing [the name of the third party contact or institution / contact details for the third party, such as an email address or a link to where data requests can be made]. Please update your statement with the missing information.

Additional Editor Comments:

1. Improve the background by updating the literature review, in particular for papers addressing the same or similar topic in Latin-america. Systematic reviews and meta-analysis have been published on the topic

2. Explain the reasons for an invalid RUN o lack of data of enrollment

3. Provide further details on the data source; why you coded the conditions? Those are not register directly there?

4. It is not clear the procedure for comorbidities, in particular what this means: Therefore, we included in the model those comorbidities with a good level of identification based on their prevalence in the sample

5. Please clarify in lines 202 & 203: those individuals were newly enrolled on that period? Are the total number of individuals enrolled? How this is a sample if you are including all?

6. In table 1 will be useful to add a column for those that tested positive

Reviewers' comments:

Reviewer's Responses to Questions

**Comments to the Author**

1. Is the manuscript technically sound, and do the data support the conclusions?

Reviewer #1: Partly

Reviewer #2: Yes

2. Has the statistical analysis been performed appropriately and rigorously? 

Reviewer #1: Yes

Reviewer #2: Yes

3. Have the authors made all data underlying the findings in their manuscript fully available?

Reviewer #1: Yes

Reviewer #2: Yes

4. Is the manuscript presented in an intelligible fashion and written in standard English?

Reviewer #1: Yes

Reviewer #2: Yes

5. Review Comments to the Author

Reviewer #1: The authors present a study aimed to evaluate the association between demographic and clinical characteristics with severe complications by SARS-CoV-2 infection, using a retrospective population-based cohort from Chile (n=44,674).

The article is well written but, in my opinion, it requires significant changes. My specifics comments are below.

Major Comments

I consider that the document lack of a clearly defined study population and study outcomes.

Given that authors aimed to evaluate complications by SARS-CoV-2 infection, it is not entirely clear why no-COVID-19 subjects were also analyzed. Additionally, considering that Ancora Network population is not representative of Chilean population, it is not evident why did the authors decided to analyze these specific inhabitants. Moreover, the terms "any hospital admission", "hospital admissions due to a COVID-19", "ICU admission" and "ICU admission due to COVID-19" are mentioned along the manuscript, making it hard to follow.

In this sense, it is important that authors bring context regarding the selection of the study population and decide to select or not only COVID-19 subjects. In the case that they decided to analyze no-COVID-19 persons, it is necessary to include an outcome regarding SARS-CoV-2 infection and develop a subgroup analysis considering this outcome.

It is also needed that authors define and homologate the study outcomes along the manuscript.

Given that the findings of this study are previously reported, in my opinion the novelty of the research objective was not entirely clear.

Minor Comments

-Introduction:

As mentioned before please include a brief explanation of the population origin. A map with location of the three primary healthcare centers and the districts may be helpful. To bring context to readers, the introduction may also include information regarding healthcare insurance, FONASA, Ancora Network and COVID-19 epidemiological surveillance in Chile.

-Materials and Methods:

Please describe the calculation of follow up time and the definition of COVID-infection (it was only by PCR? Clinical definition? Antigen Rapid test was not used for epidemiological surveillance?). As above-mentioned and similar whti COVID-19 death, definitions of main outcomes including hospitalization and UTI may be only restricted to COVID-19 related. A combination of the three outcomes may be also considered. Other variables that may be helpful may be obesity, chronic kidney disease, wave of infection, mechanical ventilation, and cause of encounters with a physician or a nurse. The subgroup analysis may be sustained in background and objectives.

-Results and Discussion:

A flowchart showing the selection steps may be included as principal figure.

Please present results regarding follow-up time.

Figure 1: please also add a curve showing the COVID-19 cases.

Table 1 and S5 table: please include totals in the first row and present statistical test comparing groups.

Table 2: add the variable "number of COVID-19 infections ".

Please correct the term "prevalence" in Line 176.

Include as limitation of the study the lack of information regarding clinical conditions control such as of diabetes and hypertension.

-Subsidiary aspects

Specify the A (lowest income), B ,C , and D (highest income) categories for FONASA variable in all tables

Reviewer #2: Interesting study

In the summary it says: "Among the 44,674 people, 460 (1.03%) had a hospital admission, 217(0.49%) had an ICU admission and 533(1.19%) died due to COVID-19," which does not apply completely with the results. I suggest writing it as it is in the results... 217 of them were also admitted to the ICU (xxx%). "In addition, the death toll must be changed from 533, which corresponds to the total cohort, to 89, which are those who died due to COVID-19."

In the summary it says: "Hypertension and diabetes pre-existing conditions … " I would eliminate "pre-existing conditions" because it is obvious that they are pre-existing.

The expression "any hospital admission" is also not clear; "hospital admission" would be better.

Also, in the summary it says: "from higher-income", it should say "from higher-income countries".

Among the authors' workplaces appears: "(6) (School of Computing and Information Systems, University of Melbourne)", however none of the authors have that membership.

It would be useful to have a table comparing the risk factors of all COVID patients with those who die from COVID, not just the entire cohort as shown in table 1.

In Sociodemographic factors it says: "People 70 years or older had 1.51 times (HR 1.51; 95% CI 1.42 to 1.62) and 38% 240 (HR 1.38; 95% CI 1.24 to 1.53) higher risk ..". I think it is better to keep the same numerical expression format, this is 1.51 times, without mixing it with percentages.

It says: "People 70 years or older had 1.86 times (HR 1.86; 95% CI 1.68 to 2.06) higher risk of dying due to a COVID-19 … than those aged 54 years or younger." However, the basis of comparison according to table 2 is with ages between 18 to 34.

I

t says: "Compared with male sex, females were 0.96, 0.94 and 0.68 times less likely .. l", the confidence intervals are missing.

The title of table number 2 says: "Hazard Ratio of the risk for any hospital admission or an ICU admission due to …… ." "COVID deaths" must be included

In Clinical factors it says: "The number of frequently dispatched drugs was associated with a higher risk of hospital and ICU admissions … " Must include numbers.

6. PLOS authors have the option to publish the peer review history of their article (what does this mean?). If published, this will include your full peer review and any attached files.

Reviewer #1: No

Reviewer #2: **Yes: **Rodrigo Gil

---

## [Author Response · Author response to Decision Letter 0]

13 Aug 2024

Thank you for the opportunity to respond to the peer reviewers’ comments. We hope we have addressed all the comments. We trust the changes made improved the manuscript. 

Our responses are included on a uploaded document at the end of this pdf. Pages and lines cited in the reponces are related to the manuscript with track-changes.

---

## [Decision Letter · Decision Letter 1]

24 Sep 2024

PONE-D-24-13634R1Association between demographic, clinical characteristics and severe complications by SARS-CoV-2 infection in a community-based healthcare network in ChilePLOS ONE

Dear Dr. Leniz,

Thank you for submitting your manuscript to PLOS ONE. After careful consideration, we feel that it has merit but does not fully meet PLOS ONE’s publication criteria as it currently stands. Therefore, we invite you to submit a revised version of the manuscript that addresses the points raised during the review process.

Please address the comments made by reviewer 1 

We look forward to receiving your revised manuscript.

Kind regards,

Juan Pablo Gutierrez

Academic Editor

PLOS ONE

Journal Requirements:

Reviewers' comments:

Reviewer's Responses to Questions

**Comments to the Author**

1. If the authors have adequately addressed your comments raised in a previous round of review and you feel that this manuscript is now acceptable for publication, you may indicate that here to bypass the “Comments to the Author” section, enter your conflict of interest statement in the “Confidential to Editor” section, and submit your "Accept" recommendation.

Reviewer #1: (No Response)

Reviewer #2: All comments have been addressed

2. Is the manuscript technically sound, and do the data support the conclusions?

Reviewer #1: Partly

Reviewer #2: (No Response)

3. Has the statistical analysis been performed appropriately and rigorously? 

Reviewer #1: Yes

Reviewer #2: (No Response)

4. Have the authors made all data underlying the findings in their manuscript fully available?

Reviewer #1: Yes

Reviewer #2: (No Response)

5. Is the manuscript presented in an intelligible fashion and written in standard English?

Reviewer #1: Yes

Reviewer #2: (No Response)

6. Review Comments to the Author

Reviewer #1: The authors had made changes improving considerably their manuscript. However, we still have the following suggestions:

-Please add information regarding COVID-19 epidemiological surveillance in Chile. Given that only PCR information was analyzed, it is important to clarify why Antigen Rapid Test data was not considered. COVID-19 epidemiological surveillance in Chile was only based on PCR? Was Antigen Rapid Test data not available for analysis? If the last case is true, this lack of availability may be pointed as a very important limitation of the report.

- The last paragraph in the introduction section benefits from a general aim that encompasses all the analysis and subgroups analysis carried out. This aim may be split into specific objectives.

- The methodology section requires an explanation regarding calculation of follow up time.

- Table 2: please revise the title which started as "Fig 4". Is the "n=42,588" correct? It is different from "n=44,674" on table 1. If the model is considering COVID-19 and non-COVID-19 cases, why is not possible to calculate HR for COVID-19 infection as outcome?

- The lack of availability of other variables such as obesity, chronic kidney disease, wave of infection, mechanical ventilation may be included into the study limitations.

-Specify the A (lowest income), B, C , and D (highest income) categories for FONASA variable in all supplementary tables and figures (S1_Table.docx, S4_Fig.docx, S5_Table.docx, S7_Table.docx).

Reviewer #2: (No Response)

7. PLOS authors have the option to publish the peer review history of their article (what does this mean?). If published, this will include your full peer review and any attached files.

Reviewer #1: No

Reviewer #2: **Yes: **RODRIGO GIL DIB

---

## [Author Response · Author response to Decision Letter 1]

6 Nov 2024

Thank you for the opportunity to respond to the peer reviewers’ comments. We hope we have addressed all the comments. Our responses are included below.

Journal Requirements:

Apologies for missing this error. We removed the following references that were reported as retracted:

• Giannouchos T V. et al. doi:10.1183/13993003.02144-2020 (Page 23, lines 536-540 tracked-changes version)

• Risk factors for mortality in patients with Coronavirus Disease 2019 (COVID-19) in Bolivia: An analysis of the first 107 confirmed cases (Page 24, lines 560-562 tracked-changes version)

And we updated the following references:

• Treskova-Schwarzbach M, et al. 2021. doi:10.2139/SSRN.3791433. In this case, we included the published version, as the previous one was a pre-print version. (Page 21, lines 472-473 tracked-changes version)

• INE. Resultados Censo 2017. 2019. In this case, we added the url and date of access. (Page 22. Lines 512-513)

Review Comments to the Author

Reviewer #1: 

2) The authors had made changes improving considerably their manuscript. 

Thank you for this supportive comment.

However, we still have the following suggestions:

3) Please add information regarding COVID-19 epidemiological surveillance in Chile. Given that only PCR information was analyzed, it is important to clarify why Antigen Rapid Test data was not considered. COVID-19 epidemiological surveillance in Chile was only based on PCR? Was Antigen Rapid Test data not available for analysis? If the last case is true, this lack of availability may be pointed as a very important limitation of the report.

Thank you for this comment. Antigen Rapid Test was available for surveillance in Chile but not from the beginning of the Pandemic. It was considered in the definition for a COVID-19 confirmed case only from 2021. The National surveillance data included all confirmed and reported COVID-19 cases, but case definition changed was not consistent across this period. Also, as we explain later, confirmed cases do not represent all COVID-19 cases during this period and less severe or asymptomatic cases are more likely to be missed than more severe cases. We agree this was not clearly explained. We included the following sentence in the method and limitation sections:

“The national COVID-19 PCR surveillance data contained information on all COVID-19 confirmed cases in Chile during the study period, with included PCR and Antigen Rapid Test confirmed cases”. (Page 5, lines 102-104 in tracked-changes version)

“Nevertheless, including only people with a positive COVID-19 PCR or Antigen Rapid Test would have sub estimate the number of people infected during this period” (Page 20, line 427 in tracked-changes version)

4) The last paragraph in the introduction section benefits from a general aim that encompasses all the analysis and subgroups analysis carried out. This aim may be split into specific objectives.

Thank you for this suggestion. We made the following changes:

“We aimed to evaluate the association between sociodemographic, clinical factors and the risk of COVID-19 complications among adults from a community-based healthcare network in Chile. We evaluated the association between sociodemographic, clinical factors and the risk of hospital admissions, admissions to ICU and death due to a COVID-19 infection, and estimated the effect by sex and COVID-19 infection in these associations.” (Page 5, lines 89-93 in tracked-changes version)

5) The methodology section requires an explanation regarding calculation of follow up time.

Thank you for this suggestion. We added the following sentence in the analysis section:

“We calculated the follow-up time based on the date the participant was registered in the healthcare centre or the study starting date (1st January 2020) whichever was later, and the date of the event (hospital admission or death) or the end of the study period (31st December 2021) whichever was first.” (Page 9, lines 192-194 in tracked-changes version)

6) Table 2: please revise the title which started as "Fig 4". 

Thank you for noticing this error. We added the space between the title of figure 4 and table 2. (page 16, line 336 in tracked-changes version)

7) Is the "n=42,588" correct? It is different from "n=44,674" on table 1. 

Thank you for raising this issue. As the variable “FONASA” has 2082 missing values, those participants the model removes those participants from the analysis. As we indicated in page 10 line 213, missing values were excluded from the model as the proportion was small (<5%).

Regardless, 4 participants were wrongly coded as having a cero following time and therefore removed from the Cox regression model. We rectified that error in the data and ran the models again. This did not change the results, so tables were not amended except for the n that was changed to 42,592 in table 2; and 18,496 and 24,096 for males and females respectively in table S7.

8) If the model is considering COVID-19 and non-COVID-19 cases, why is not possible to calculate HR for COVID-19 infection as outcome?

Thank you for this question. We can calculate the HR for COVID-19 infection indeed. What we meant previously is that including the variable COVID-19 infection in models for any hospital admission or death due to a COVID-19 infection was not possible as all participants with a hospital admission due to COVID-19 or COVID-19 death were COVID-19 infected patients as well. Apologies if our former explanation was not clear. 

We explicitly decided not to include COVID-19 infection as an additional outcome in our study because it was not the aim of the study to estimate the risk of COVID-19 infection but the risk of COVID-19 complications. Also, we cannot be certain all COVID-19 cases in the community were properly identified, except in hospital or decedents. It is very unlikely that a COVID-19 case was not identified in hospital or mortality records, as patients were regularly tested in these settings. But it is very likely COVID-19 case were not identified in the community setting. Therefore, calculating the risk of a COVID-19 infection at a population level would underestimate the risk, as the COVID-19 infections identified in our data are likely to represent more severe cases.

9) The lack of availability of other variables such as obesity, chronic kidney disease, wave of infection, mechanical ventilation may be included into the study limitations.

We appreciate this suggestion. We added the following sentence in the strengths and limitation section:

“We were not able to include all pre-existing chronic conditions associated with COVID-19 infection in the literature, such as obesity, chronic kidney disease, symptoms, diabetes and hypertension level of control, or factors such as mechanical ventilation or wave of infection due to the lack of structured and systematic recording in primary and hospital healthcare records.” (Page 19, lines 419-421)

10) Specify the A (lowest income), B, C , and D (highest income) categories for FONASA variable in all supplementary tables and figures (S1_Table.docx, S4_Fig.docx, S5_Table.docx, S7_Table.docx).

Thank you for this comment. We amended the tables and figures as requested.

---

## [Editor Report · Decision Letter 2]

11 Nov 2024

Association between demographic, clinical characteristics and severe complications by SARS-CoV-2 infection in a community-based healthcare network in Chile

PONE-D-24-13634R2

Dear Dr. Leniz,

We’re pleased to inform you that your manuscript has been judged scientifically suitable for publication and will be formally accepted for publication once it meets all outstanding technical requirements.

Kind regards,

Juan Pablo Gutierrez

Academic Editor

PLOS ONE
---

## [Editor Report · Acceptance letter]

15 Nov 2024

PONE-D-24-13634R2 

PLOS ONE

Dear Dr. Leniz, 

I'm pleased to inform you that your manuscript has been deemed suitable for publication in PLOS ONE. Congratulations! Your manuscript is now being handed over to our production team.

Kind regards, 

on behalf of

Dr. Juan Pablo Gutierrez 

Academic Editor

PLOS ONE